# 4D Gaussian Splatting with Scale-aware Residual Field and Adaptive Optimization for Real-time rendering of temporally complex dynamic scenes

Submission Id: 4096

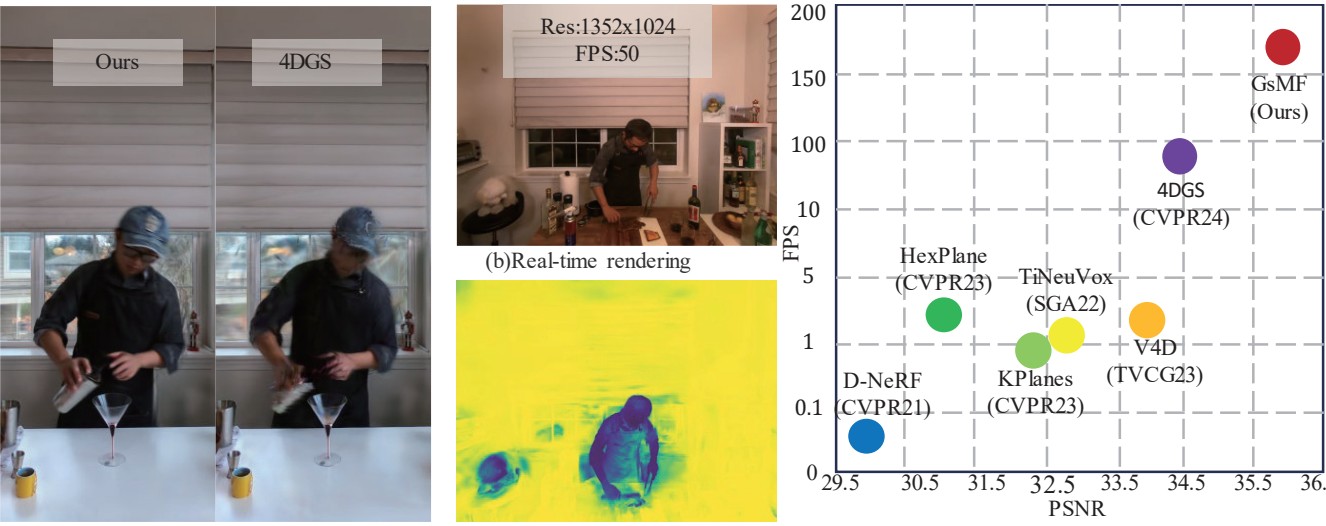

**Figure 1: Performance comparison with previous SOTA[5–7, 9, 31, 45]. Our approach achieves higher-quality reconstruction in temporally complex scenes (a) while maintaining real-time rendering (b), with a certain improvement in performance(c). Additionally, we achieve dynamic scene segmentation without any prior information (c).**

## ABSTRACT

Reconstructing dynamic scenes from video sequences is a highly promising task in the multimedia domain. While previous methods have made progress, they often struggle with slow rendering and managing temporal complexities such as significant motion and object appearance/disappearance. In this paper, we propose SaRO-GS as a novel dynamic scene representation capable of achieving real-time rendering while effectively handling temporal complexities in dynamic scenes. To address the issue of slow rendering speed, we adopt a Gaussian primitive-based representation and optimize the Gaussians in 4D space, which facilitates real-time rendering with the assistance of 3D Gaussian Splatting. Additionally, to handle temporally complex dynamic scenes, we introduce a Scale-aware Residual Field. This field considers the size information of each Gaussian primitive while encoding its residual feature and aligns with the self-splitting behavior of Gaussian primitives. Furthermore, we propose an Adaptive Optimization Schedule, which assigns different optimization strategies to Gaussian primitives based on their distinct temporal properties, thereby expediting the reconstruction of dynamic regions. Through evaluations on monocular and multi-view datasets, our method has demonstrated state-of-the-art performance.

## CCS CONCEPTS

• **Computing methodologies → Rendering**.

## KEYWORDS

Real-time rendering,Dynamic scene reconstruction

## 1 INTRODUCTION

The reconstruction of dynamic scenes is pivotal for immersive imaging, driving advancements in various multimedia technologies such as VR, AR, and metaverse. Rendering dynamic scenes from any time, position, and viewpoint is crucial for enhancing user experiences with multimedia products, like free-viewpoint video and bullet-time effects. Our objective is to reconstruct a continuous 4D space from a discrete temporal video sequence. However, this endeavor faces several challenges. Firstly, the reconstruction quality acts as

a bottleneck for widespread adoption, requiring accurate capture of spatial dimensions and temporal variations in dynamic scenes. Additionally, there's a growing demand for real-time interaction in multimedia products to boost user engagement, highlighting the importance of achieving real-time rendering. Nevertheless, existing methods struggle to achieve both high-quality reconstruction and real-time rendering simultaneously, precisely the issue our approach aims to tackle.

Recent advancements in dynamic scene reconstruction have been achieved through methods based on NeRF [23] and 3DGS [14]. NeRF employs an implicit field to model static scenes and achieves photo-realistic view synthesis. Many extensions of NeRF to dynamic scenes either utilize deformation fields and canonical fields to model the motion of objects relative to canonical frames over time[10, 20, 27, 31, 35, 38], or decompose the 4D volume into spatial-only and spatial-temporal spaces[5, 7, 19, 34], representing space through combinations of dimensionally reduced features. While significant progress has been made in rendering quality, these methods face a significant disadvantage in rendering speed. The emergence of 3DGS has enabled real-time rendering of dynamic scenes. Some methods [13, 18, 45–47] have attempted dynamic scene modeling based on 3DGS. However, they either struggle to model temporally complex scenes such as object appearances and disappearances [13, 45, 46] or overlook the spatiotemporal information in the scene [18, 47], resulting in disadvantages when dealing with temporally complex dynamic scenes.

To address the aforementioned challenges, we propose SaRO-GS, aiming to achieve real-time rendering while maintaining high-quality reconstruction of temporally complex dynamic scenes. SaRO-GS comprises a set of Gaussian primitives in 4D space and a Scale-aware Residual Field. Each Gaussian receives a unique optimization schedule based on its distinct temporal properties through an Adaptive Optimization strategy. To address the issue of slow rendering speeds, Gaussian primitives in 4D space can be projected to 3D based on their temporal properties and residual features obtained from the Scale-aware Residual Field. Then we can achieve real-time rendering leveraging the fast differentiable rasterizer introduced by 3DGS. For high-quality modeling of temporally complex scenes, we employ the following strategies: Firstly, each 4D Gaussian primitive possesses temporal properties, including temporal position and lifespan. The lifespan allows us to model the appearance and disappearance of objects in dynamic scenes, while the temporal position of Gaussians spans the entire temporal range, rather than being fixed at frame 0 as in previous methods. Additionally, we incorporate scale information of Gaussian primitives into the Residual Field to accommodate their ellipsoidal nature. By encoding the region that the Gaussian primitives occupy rather than just their position, we ensure accurate feature extraction and align with the self-splitting behavior of Gaussian primitives. Thirdly, we introduce an Adaptive Optimization strategy, where unique optimization strategies are assigned to each Gaussian primitive based on its temporal properties, facilitating faster reconstruction of dynamic regions.

We extensively evaluated our approach on monocular and multi-view dynamic scene datasets, comprising both real and synthetic scenes. Both quantitative and qualitative results demonstrate that

our method achieves high-quality rendering in real time and effectively handles temporal complexities in dynamic scenes. Our contributions are summarized below.

- We propose a Scale-aware Residual Field, incorporating the scale information of Gaussian primitives. This results in a more precise spatiotemporal representation, considering Gaussian primitives' ellipsoidal nature and self-splitting behavior.
- We introduce an Adaptive Optimization strategy, assigning unique optimization schedule to Gaussians based on their unique temporal properties, enhancing the reconstruction of dynamic areas.
- Our SaRO-GS excels in managing temporally complex scenarios, delivering state-of-the-art performance in both the reconstruction quality and rendering speed. It achieves an 80x rendering speed improvement compared to NeRF-based methods as shown in Fig. 1. SaRO-GS is versatile, applicable to both monocular and multi-view scenarios and can also achieve dynamic scene segmentation without any prior knowledge.

## 2 RELATED WORK

### 2.1 Neural Scene Representations

In recent years, there has been widespread interest in the representation of static scenes based on Neural Radiance Fields. NeRF[23], as a representative work, adopts a purely implicit approach, modeling static scenes as a radiance field and synthesizing images through volume rendering. The photo-realistic view synthesis capability of NeRF has inspired a series of works across various domains, including enhancing rendering quality[3, 39, 40, 51, 53],different camera trajectories[4, 43],sparse inputs[2, 25],accelerating training and rendering[8, 11, 12, 24, 32–34, 36, 48, 52], as well as human body reconstruction[29, 30, 44, 50] and large-scale scene modeling[22, 37], among others.

Recently, there has been a breakthrough in high-quality view synthesis and real-time rendering with 3D Gaussian Splatting[14] and relate works[16, 49, 54, 55], garnering significant attention in the scene reconstruction field.

### 2.2 Dynamic Scene Representation Based on NeRF

Expanding static scene representation to dynamic scenes is not a simple task. Some methods[10, 20, 27, 31, 35, 38] have made progress based on deformation fields, modeling the entire scene as a canonical field and a deformation field. They use the deformation field to represent the association between sampled points under different frames and the static canonical. Other methods[5, 7, 19, 34] reduce the dimensionality of the 4D space by decomposing it into a set of planar grids, projecting the 4D sampled points onto planes to obtain corresponding features. This approach effectively models temporal correlations through spatiotemporal planes. Furthermore, some methods[17, 42] adopt a streaming strategy to model residuals between adjacent frames, which is suitable for real-time transmission and decoding. However, these rendering approaches based on NeRF requires dense sampling along rays during rendering, limiting the possibility of real-time rendering.

## 2.3 Dynamic Scene Representation Based on 3D Gaussian

There are some concurrent works based on 3D Gaussian representations for dynamic scenes. [21] uses an online strategy to model dynamic scenes frame by frame, [45] uses a Hex-planes to model the changes of Gaussian primitives over sampling time. Both of them struggle to handle temporal complexities, such as significant motion and object appearances/disappearances. [47] introduces a time dimension to Gaussian, enabling 4D Gaussian to be decomposed into a conditional 3D Gaussian and a marginal 1D Gaussian. In comparison, we propose an Scale-aware Residual field to model the residual of Gaussian primitives projected from 4D to 3D, employing an explicit-implicit blending approach that better incorporates spatiotemporal correlations. Other methods like [18] are not suitable for monocular scenarios, and [46] cannot be utilized in multi-view scenes. In contrast, our approach, validated through experiments, demonstrates promising results in both single-view and multi-view scenarios.

## 3 PRELIMINARY

### 3.1 3D Gaussian Splatting

Given a set of input images of a static scene along with their corresponding camera parameters, 3D Gaussian Splatting (3DGS) initiates the reconstruction of the static scene from an initial point cloud, employing 3D Gaussians as primitives. This approach enables high-quality real-time novel view synthesis.

In 3DGS, each Gaussian primitive encompasses a set of attributes, including 3D position $\mu_{3d}$, opacity $\alpha$, and covariance matrix $\Sigma$. A 3D Gaussian $\mathcal{G}$ can be represented as:

$$G(x) = e^{-\frac{1}{2}(x-\mu)^T \Sigma^{-1}(x-\mu)} \tag{1}$$

For optimization convenience, 3DGS employs a scaling matrix $S$ and a rotation matrix $R$ to represent covariance, stored as a 3D vector $s$ for scaling and a quaternion $q$ for rotation.

$$\Sigma = RSS^T R^T \tag{2}$$

Additionally, 3DGS utilizes SH coefficients to represent view-dependent color.

Based on the fast differentiable rasterizer implemented by 3DGS, we can achieve rapid image rendering through Gaussian Splatting. To get the rendering images from a given viewpoint, we should first project 3D Gaussian primitives to 2D. Specifically, for a given viewpoint transformation matrix $W$ and a projection matrix $K$, we can obtain the covariance and the position in 2D space:

$$\Sigma^{2D} = (JW\Sigma W^T J^T)_{1:2,1:2}$$
$$\mu^{2D} = K\left(\frac{W\mu}{(W\mu)_z}\right)_{1:2} \tag{3}$$

where $J$ is the Jacobian of the affine approximation of the projetive transformation. And we can get the 2D Gaussian $\mathcal{G}^{2d}$ based on Eq. 1.

After sorting the Gaussian primitives in 2D space based on depth, we can obtain the color of the specified pixel in the image:

$$c(x) = \sum_{i=1}^{N} c_i \alpha_i \mathcal{G}_i^{2D} \prod_{j=1}^{i-1}(1 - \alpha_j \mathcal{G}_j^{2D}(x)) \tag{4}$$

Here, $c_i$ represents the view-dependent color obtained by combining the SH coefficients of $\mathcal{G}_i$ with the viewing direction.

## 3.2 4D Volume Representation Based on Hex-planes

Previous works modeling dynamic scenes using plane field encoders mostly employed hexplanes $P$, which encompass spatial-only planes $P_{so} = \{P_{x,y}, P_{y,z}, P_{x,z}\}$ and spatiotemporal planes $P_{st} = \{P_{x,t}, P_{y,t}, P_{z,t}\}$. Each plane $P_{i,j}$ in $P$ is a $M \times N \times N$ two-dimensional grid, where $M$ represents the feature dimension and $N$ represents the spatiotemporal resolution of the grid. For a 4D sample point $q = (x,y,z,t)$, we perform interpolation based on its projected coordinates on the six grids to obtain the corresponding feature for this point:

$$f(q) = \prod_{i,j \in C} \psi_{bi}(\pi_{i,j}(q); P_{i,j})$$
$$C = \{(x, y), (x, z), (x, t), (y, z), (y, t), (z, t)\} \tag{5}$$

Here, $f(q)$ is an $M$-dimensional feature, $\psi_{bi}$ represents bilinear interpolation, and $\pi_{i,j}(q)$ denotes the projected coordinates of sample point $q$ on $P_{i,j}$.

## 4 METHOD

In this section, we initiate with the presentation of overall pipeline of SaRO-GS in Sec. 4.1. Subsequently, we explore the Scale-aware Residual Field in Sec. 4.2, with a particular emphasis on its consideration of scale information for Gaussian primitives. Following this, we elaborate on the Adaptive Optimization strategy employed in Sec. 4.3. Finally, we delineate the loss function and regularization terms utilized in our approach in Sec. 4.4.

### 4.1 Representation of SaRO-GS

To represent a dynamic scene, we utilize a set of Gaussian primitives $\mathcal{G}^{4D}$ in 4D space alongside Scale-aware Residual Field $\mathcal{M}$, as shown in Fig. 2(a). Each 4D Gaussian primitive $\mathcal{G}_i^{4D}$ possesses a temporal position $\tau_i$, it is learned alongside its 3D position (x,y,z), forming a 4D location $\mu^{4D} = (x, y, z, \tau)$. Together with the initial attributes $\Sigma^{4D}$, $c^{4D}$ and $\alpha^{4D}$, a 4D Gaussian primitive and its residual feature $f$ can be represented as follows:

$$\mathcal{G}^{4D} = (\mu^{4D}, \Sigma^{4D}, c^{4D}, \alpha^{4D}), \tag{6}$$

$$f = \mathcal{M}(\mathcal{G}^{4D}). \tag{7}$$

$\Sigma^{4D}$, $c^{4D}$ and $\alpha^{4D}$ respectively represent the initial covariance, color, and opacity of the Gaussian primitive in 4D space. Similar to 3D Gaussian, we employ quaternion rotation $q^{4D} = (q_a, q_b, q_c, q_d)$ and scaling vectors $s^{4D} = (s_x, s_y, s_z)$ to represent covariance, and utilize SH (Spherical Harmonics) coefficients to depict view-dependent color.

To address complex temporal scenarios such as object appearance and disappearance, each Gaussian primitive should have a lifespan to indicate how long it can survive in the temporal domain. In order to effectively integrate the Scale-aware Residual Field $\mathcal{M}$ with our 4D Gaussian primitives and leverage the spatiotemporal characteristics of $\mathcal{M}$, we employ a tiny MLP $\mathcal{F}_w$ to perform

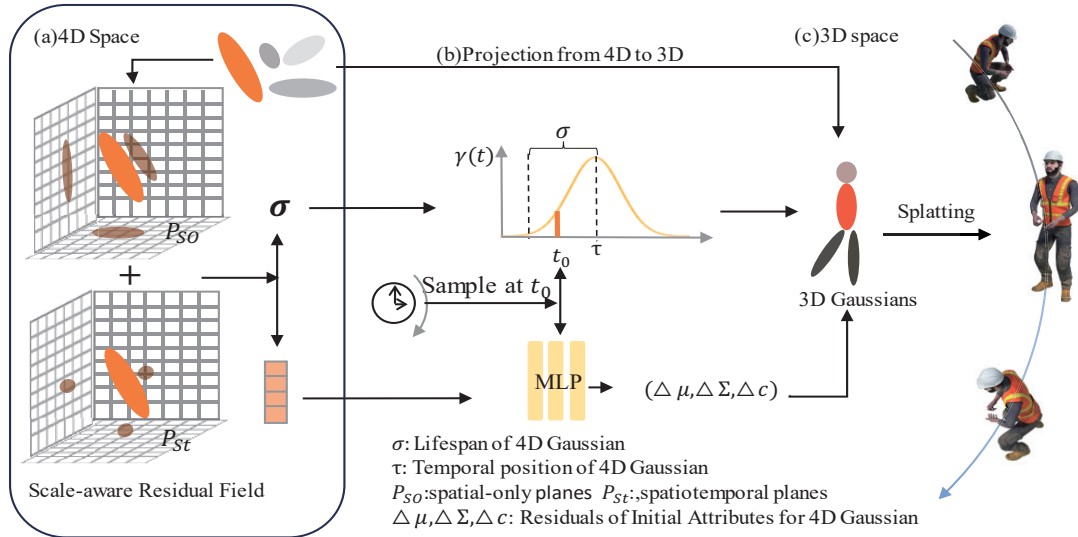

**Figure 2: The overall pipeline of SaRO-GS. (a)In 4D space, we simultaneously optimize a set of 4D Gaussian primitives and a scale-aware Residual Field $\mathcal{M}$. When combined with $\mathcal{M}$, each Gaussian primitive generates a residual feature and a lifespan $\sigma$. They both represent the temporal characteristics of the Gaussian primitive. (b)Given a sampling time $t_0$, we can compute the survival status $g(t_0)$ of the Gaussian and decode the residual feature of the Gaussian at time $t_0$ using an MLP, yielding residual of atteibutes. Finally, we combine these residuals with the initial attributes of the Gaussian in 4D space to get the 3D Gaussian representation.(c) Once we obtain the representation of the 3D Gaussian, we can generate rendered images using Gaussian Splatting.**

inference on $f_i$ and compute the lifespan $\sigma_i$ of $\mathcal{G}_i^{4D}$:

$$\sigma_i = \mathcal{F}_w(f_i) = \mathcal{F}_w(\mathcal{M}(\mathcal{G}_i^{4D})). \tag{8}$$

Therefore, in our 4D space, each Gaussian primitive $\mathcal{G}_i^{4D}$ can obtain a residual feature $f_i$ and a lifespan $\sigma_i$ through $\mathcal{M}$. $\{\mathcal{G}_i^{4D}, f_i, \sigma_i\}$ fully represents both the initial attributes and temporal characteristics of a Gaussian primitive in 4D space.

Once the sampling time $t_0$ are given, we need to project the Gaussian primitives from 4D space to 3D space, as shown in Fig. 2(b). We first need to examine whether $\mathcal{G}_i^{4D}$ still survives at the current sampling time $t_0$. We adopt a Gaussian-like state function $\gamma(t)$ to model the state of $\mathcal{G}_i^{4D}$ as it varies with the sampling time $t$:

$$\gamma_i(t) = e^{-k\frac{t-\tau_i}{\sigma_i}^2}. \tag{9}$$

Where $t$ represents the sampling time, and $\tau_i$ represents the temporal position of $\mathcal{G}_i^{4D}$. In practice, $k$ is set to 4. As the sampling time $t$ gradually moves away from the temporal position of $\mathcal{G}_i^{4D}$, $\gamma(t)$ decreases from 1. When the sampling time reaches the Gaussian lifespan $\sigma_i$, $\gamma(t)$ will decrease to 0.01, indicating that $\mathcal{G}_i^{4D}$ is nearly inactive at $t$, which means it should be invisible when projected into 3D space. So for a given sampling time $t_0$, We can utilize this state function to represent the opacity of $\mathcal{G}_i^{4D}$ in 3D space after projection:

$$\alpha_i^{3D} = \alpha_i^{4D} \times \gamma_i(t_0) \tag{10}$$

Apart from opacity, other features of $\mathcal{G}_i^{4D}$ also vary with sampling time $t$ when projected into 3D space. We can utilize a set of MLPs $\mathcal{F}_\theta$ to decode the residual feature $f_i$ of $\mathcal{G}_i^{4D}$ at the sampling time $t$, thereby obtaining the residual of the projected attributes as they vary with the sampling time $t$.

$$\Delta\mu_i(t), \Delta\Sigma_i(t), \Delta c_i(t) = \mathcal{F}_\theta(f(\mathcal{G}_i^{4D}), t - \tau_i) \tag{11}$$

$\Delta\mu_i(t), \Delta\Sigma_i(t)$ and $\Delta c_i(t)$ respectively represent the residuals of position, covariance, and color. Here we decode using $t - \tau_i$ instead of just $t$, as we aim to obtain the residual relative to the initial attribute of $\mathcal{G}_i^{4D}$ in 4D space, where $\mathcal{G}_i^{4D}$ is temporally positioned using $\tau_i$.

Therefore , we can obtain the attributes of projected $\mathcal{G}_i^{3D}$ at a given time $t_0$:

$$\mu_i^{3D} = \mu_i^{4D}[:3] + \Delta\mu_i(t_0), \tag{12}$$
$$\Sigma_i^{3D} = \Sigma_i^{4D} + \Delta\Sigma_i(t_0), \tag{13}$$
$$c_i^{3D} = c_i^{4D} + \Delta c_i(t_0). \tag{14}$$

$\mu_i^{4D}[:3]$ represents extracting the xyz components of $\mu_i^{4D}$ as the initial 3D location, and $\Delta\Sigma_i$ can be decomposed into $\Delta\Sigma_i[:3]$ and $\Delta\Sigma_i[3:]$, representing the residuals of the three-dimensional scaling vector and quaternion rotation about $\mathcal{G}_i^{4D}$, respectively. The changes to $\Sigma_i^{4D}$ and $c_i^{4D}$ are achieved by adjusting the corresponding quaternion rotation, scaling vectors, and SH coefficients.

Therefore, according to Eq. [8-14], Gaussian primitives in 4D space evolve within their lifespan as the sampling time varies, projecting into 3D space at a given sampling time. Then, based on Eq. 34, we employ 3DGS to render 3D Gaussians, obtaining rendered images from a given camera viewpoint, as shown in Fig. 2(c).

## 4.2 Scale-aware Residual Field

To fully integrate the spatiotemporal information of the scene and save computational resources, we adopt hexplanes to represent our Scale-aware Residual Field $\mathcal{M}$, which consists of spatial-only planes and spatiotemporal planes, as described in Sec. 3.2.

However, neglecting the size of Gaussian primitives and solely projecting them onto planes based on their 4D positions for feature extraction would lead to incorrect residual features. First, Gaussian

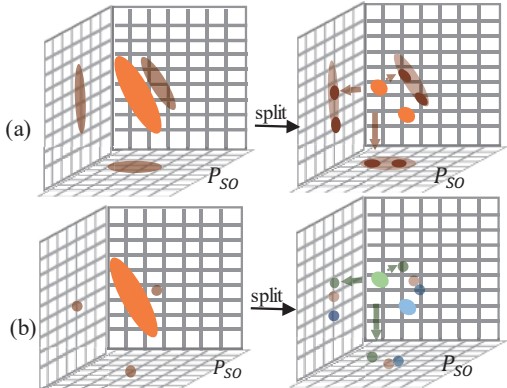

**Figure 3: The impact of scale is not taken into account in Gaussian self-splitting. (a)When size information is considered, the features of the split Gaussian remain similar to its parent Gaussian. (b)Otherwise, the split Gaussian will have features different from its parent Gaussian**

primitives can be approximated as ellipsoids. Therefore, when projecting a Gaussian primitive onto a spatial-only plane, we obtain an elliptical region instead of a single point, as in the current NeRF-based approach. Thus, the corresponding feature for a Gaussian primitive should be a combination of all the areas it occupies in the plane. Secondly, if we follow the self-splitting strategy of 3DGS and split a large Gaussian primitive into smaller ones, they would have different residual features, significantly deviating from those of their parent primitive, which contradicts our original intention, as shown in Fig. 3. So, finding an appropriate method to encode the region projected by Gaussian is crucial.

We propose a scale-aware Residual Field to address the aforementioned issue, which decompose the 4D space into three spatial-only planes $P_{so}$ and three spatiotemporal planes $P_{st}$. Given that the size of Gaussian primitives impacts their projection only within the spatial-only plane $P_{so}$, we specifically consider only employing scale-aware encoding within them, as shown in Fig. 2.

For every spatial-only planes $P_{i,j}$, we employ a MipMap stack to represent features at different spatial scales in the scene. The level 0 of the MipMap stack $P_{i,j}^0$ is a feature map with shape $M \times N \times N$ ,which has the smallest spatial scale $\ddot{s}^0$ among all levels. And remaining levels $P_{i,j}^l$ in the Mipmap stack are obtained by computing thumbnails based on the features of the previous level, where the width and height are reduced by a factor of 2 each. Taking $P_{x,y}$ as an example, the relationship between their spatial scale is as follows:

$$\ddot{s}_x^0 = \frac{\mathcal{B}_{max}^x - \mathcal{B}_{min}^x}{N} \qquad \ddot{s}_y^0 = \frac{\mathcal{B}_{max}^y - \mathcal{B}_{min}^y}{N}$$
$$\ddot{s}^{l+1} = 2 \times \ddot{s}^l \qquad\qquad l \in [0, L-1] \tag{15}$$

The variables $\mathcal{B}_{max}$ and $\mathcal{B}_{min}$ respectively represent the maximum and minimum values of the scene's bounding box and $\ddot{s}^l$ is the spatial scale of level $l$ in the MipMap stack. In practice, we only store and optimize the features in level 0 MipMap $P_{x,y}^0$, while the remaining levels are dynamically computed and generated during forward inference. This way, we possess the capability to encode features at different spatial scales within the scene.

Meanwhile, for a Gaussian primitive with a scaling s in 4D space, when projected onto the spatial-only plane $P_{x,y}$, it results in a 2D ellipse with axes $(s_x, s_y)$.Therefore, based on the projected axes of the Gaussian primitive on $P_{x,y}$ and the corresponding base spatial scale $\ddot{s}_{x,y}^0$ of the MipMap stack, we can determine the spatial scale level associated with this Gaussian primitive:

$$l^x = log2(\frac{s_x}{\ddot{s}_x^0}) \qquad l^y = log2(\frac{s_y}{\ddot{s}_y^0}) \tag{16}$$

To maintain the highest possible accuracy, we choose the minimum value among them as the final spatial level $l = min(l^x, l^y)$. So now we can obtain the two MipMap features that are closest to its spatial level:$P_{x,y}^{\lfloor l \rfloor}, P_{x,y}^{\lceil l \rceil}$, and we can obtain the embedding of the Gaussian primitive with 4D position $\mu_{4d}$ in $P_{x,y}$ as:

$$f_{x,y} = Tri\psi(\pi_{x,y}(\mu^{4D}), l; P_{x,y}^{\lfloor l \rfloor}, P_{x,y}^{\lceil l \rceil},) \tag{17}$$

Here, $\psi_{tri}$ represents trilinear interpolation in the space formed by $P_{x,y}^{\lfloor l \rfloor}$ and $P_{x,y}^{\lceil l \rceil}$. The complete expression of the scale-aware residual feature $f$ of $\mathcal{G}^{4D}$ is as follows:

$$
\begin{aligned}
f(\mathcal{G}^{4D}) &= f_{so} + f_{st} \\
&= \sum_{i,j \in C_{so}} \psi_{tri}(\pi_{x,y}(\mu^{4D}), l; P_{x,y}^{\lfloor l \rfloor}, P_{x,y}^{\lceil l \rceil},)+ \\
&\quad \sum_{i,j \in C_{st}} \psi_{bi}(\pi_{i,j}(\mu^{4D}); P_{i,j}) \\
C_{so} &= \{(x,y),(x,z),(y,z)\} \qquad C_{st} = \{(x,t),(y,t),(z,t)\}.
\end{aligned}
\tag{18}
$$

Here, $\psi_{tri}, \psi_{bi}$ represent trilinear interpolation and bilinear interpolation respectively. Through experiments, we found that summation is a more effective way to combine features in our Scale-aware Residual Field compared to others.

## 4.3 Adaptive Optimization

Due to the varying temporal locations and lifespans of Gaussian primitives in this 4D space, each Gaussian primitive is sampled with different probabilities over observed time. Dynamic primitives, in order to represent the temporal complexity of the scene, often have a smaller lifespan, resulting in a lower sampling probability compared to static primitives. Primitives with less exposure over the entire temporal domain would have smaller gradients during the backward propagation of the loss function. The gradient value is crucial in the 3DGS framework, as it needs to exceed a threshold to densify the corresponding primitive and optimize the currently imperfectly reconstructed regions. Hence, applying the same optimization and densification strategy directly to each primitive with 3DGS[14] may lead to optimization imbalance.

To tackle the aforementioned issue, we propose an Adaptive Optimization strategy, which dynamically adjusts the learning rate and densify gradient threshold for $\mathcal{G}_i^{4D}$ based on its sampling probability across the observable time range. Specifically, We can use $\gamma_i(t)$ of $\mathcal{G}_i^{4D}$ to calculate the temporal integral within the observable range, representing its sampling probability . The larger the integral, the more the Gaussian primitive's lifespan intersects with the observable range, making it more likely to be sampled.

**Definite Integral of the Time Domain Distribution.** Based on state function $\gamma_i(t)$ of $\mathcal{G}_i^{4D}$, we can compute its integral over the

time domain.

$$F(t) = P(x < t) = \int_{-\infty}^{t} e^{-k\frac{x-\tau_i}{\sigma_i}^2} dx \qquad (19)$$

$$I = F(t_{end}) - F(t_{start}) \qquad (20)$$

$F(t)$ represents the CDF (cumulative distribution function) of each Gaussian primitive. Here, $I$ denotes the definite integral over the entire time domain from $t_{start}$ to $t_{end}$, where $t_{start}$ and $t_{end}$ normalize to 0 and 1, respectively.

Since $\gamma_i(t)$ is a Gaussian-like function, it is challenging to compute precise definite integral values. Inspired by [26], we derive the approximate cumulative distribution function as:

$$Q(t) = \int_{-\infty}^{t} \frac{1}{\sqrt{2\pi}} e^{-\frac{x^2}{2}} dx = 1 - \frac{1}{e^{1+\alpha_1 t^3 + \alpha_2 t}} \qquad (21)$$

$$F(t) = \frac{\sqrt{\pi}\sigma_i}{\sqrt{k}} Q(\sqrt{2k}\frac{(t-\tau_i)}{\sigma_i}) \qquad (22)$$

Here, $\alpha_1 = 0.070565992$, $\alpha_2 = 1.5976$. Please refer to the appendix for details. Therefore, we can obtain the definite integral of Gaussian primitives over the time domain using Eq. 20 with minimal computational complexity.

**Intergral-based Per-Gaussian Optimization Schedule.** After get the temoral intergral of each Gaussian, we can dynamically adjust the learning rates and gradient thresholds for densification control on a per-primitive basis, aiming to achieve rapid reconstruction of dynamic regions:

$$\kappa_i = \kappa_{base} * \frac{I_i}{I_{max}}, \qquad lr_i = lr_{base} * \frac{I_{max}}{I_i} \qquad (23)$$

Here, $\kappa_i$ and $lr_i$ respectively represent the densification threshold and the learning rate of $\mathcal{G}_i^{4D}$, while $I_i$ and $I_{max}$ respectively denote the timeporal integral of $\mathcal{G}_i^{4D}$ and the maximum time-domain integral among all Gaussian primitives. We adjust its densification threshold each time densification control is required. Additionally, for the learning rate, we dynamically adjust it every 50 iterations based on Eq. 23, involving parameters related to 4D position, scaling, rotation, and zeroth-order SH coefficients of $\mathcal{G}_i^{4D}$.

## 4.4 Loss Function

**Regularization term for scaling residuals:** When the sampling time $t$ equals the temporal position of $\mathcal{G}_i^{4D}$, we aim for minimal variations in the attributes of Gaussian primitives projected into 3D space compared to their initial values in 4D space. Excessive reliance on attribute residuals during the projection process may neglect the optimization of their initial values. Additionally, we strive to minimize the disparity between the initial scaling values of Gaussian primitives in 4D space and their values in 3D space. This ensures that the Scale-aware Residual Field can effectively integrate accurate scale information. To achieve this goal, we propose a regularization term $\mathcal{L}_{SR}$ concerning scaling residuals:

$$\mathcal{L}_{SR}(\mathcal{G}^{4D}) = \frac{1}{n} \sum_i ||\Delta\Sigma_i(\tau_i)[:3]||_2 \qquad (24)$$

Here, $\Delta\Sigma_i(\tau_i)[:3]$ represents the residuals of scaling of $\mathcal{G}_i^{4D}$ obtained during the projection process when the sampling time is $\mathcal{G}_i^{4D}$'s temporal position $\tau_i$, where $n$ is the total number of Gaussian primitives in the current 4D space.

**Table 1: Quantitative results on the monocular synthesis dataset D-NeRF. FPS is measured at $400 \times 400$ . † denotes a dynamic Gaussian method. The evaluation of [1] is conducted at a resolution of 800, while the remaining methods are evaluated at 400**

| Method | PSNR(dB)↑ | SSIM↑ | LPIPS↓ | FPS↑ | Train Time |
|---|---|---|---|---|---|
| D-NeRF[31] | 29.67 | 0.95 | 0.07 | 0.06 | 48h |
| KPlanes-hybrid[7] | 32.36 | 0.96 | - | 0.97 | 52m |
| TiNeuVox-B[6] | 32.67 | 0.97 | 0.04 | 1.5 | 28m |
| V4D[9] | 33.72 | 0.98 | 0.02 | 2.08 | 6.9h |
| HexPlane[5] | 31.04 | 0.97 | 0.04 | 2.5 | 11m 30s |
| 4DGS[45]†[1] | 34.05 | 0.98 | 0.02 | 82 | 20m |
| 4DGS-Realtime[47]† | 34.09 | 0.98 | - | - | - |
| Ours | 36.13 | 0.98 | 0.01 | 182.29 | 45m |

**Total Loss Function** Following [14], we use the loss between the rendered image and the ground truth image, which includes an $\mathcal{L}_1$ term and a $\mathcal{L}_{D-SSIM}$ term. Combined with our regularization terms, the overall loss function is formulated as:

$$\mathcal{L} = (1 - \lambda_1)\mathcal{L}_1 + \lambda_1 \mathcal{L}_{D-SSIM} + \lambda_2 \mathcal{L}_{SR} \qquad (25)$$

The settings for $\lambda_1$ and $\lambda_2$ can be referred to in Sec. 5.

## 5 IMPLEMENTATION DETAILS

We implemented our work using the PyTorch[28] framework and open-source code based on 3DGS. We utilized the nvidiffrast[15] library to compute the MipMap stack in the Scale-aware Residual Field, ensuring computational efficiency. We use Adam optimizer and retained certain implementations from 3DGS, including the fast differentiable rasterizer, hyperparameters, and opacity reset strategy. In the loss function Eq. 25, we set $\lambda_1$ to 0.2 and $\lambda_2$ to 0.8. Detailed hyperparameter settings can be found in the appendix. We conducted training and testing of our model using a single RTX 3090.

To enhance our rendering speed, we adopted a lossless baking strategy for the model during rendering. Since the features extracted from the Scale-aware Residual Field are independent of the sampling time, we can pre-compute the features for each Gaussian primitive in the 4D space. Thus, during rendering, the conversion of Gaussian primitives from 4D space to 3D space incurs only the overhead of MLP inference. Additionally, during rendering, for a given sampling time $t_0$, we filter out points where the survival status $\gamma(t_0) < 0.001$. These filtered points indicate that they are inactive and invisible at time $t_0$), thereby reducing the computational overhead during MLP Decoder and Splatting. After testing, our baking strategy has enabled us to double our rendering speed.

## 6 EXPERIMENTS

### 6.1 Datasets

**Synthetic Dataset.** We chose D-NeRF[31] as our evaluation dataset for monocular scenes. D-NeRF is a monocular synthetic dataset consisting of eight scenes with large-scale movements and real non-Lambertian material dynamic objects, which imposes a challenge on model performance.

**Real-world Datasets.** We selected Plenoptic Video dataset to evaluate our performance in multi-view real dynamic scenes. Plenoptic Video dataset consists of six real-world scenes, each captured by 15-20 cameras. Each scene in the dataset encompasses complex

**Figure 4: Qualitative result on the D-NeRF dataset. It can be observed that our method outperforms others in the reconstruction of details.**

**Table 2: Quantitative results on Plenoptic Video dataset. FPS is measured at $1352 \times 1014$. † denotes a dynamic Gaussian method.[1] excludes the Coffee Martini scene.**

| Method | PSNR(dB)↑ | DSSIM↓ | LPIPS↓ | FPS |
|--------|-----------|--------|--------|-----|
| KPlanes-hybrid[7] | 31.63 | - | - | 0.3 |
| Mix-Voxels-L[41] | 31.34 | 0.017 | 0.096 | 16.7 |
| NeRFPlayer[35] | 30.69 | 0.034 | 0.111 | 0.045 |
| HyperReel[1] | 31.1 | 0.037 | 0.096 | 2.00 |
| StreamRF[17] | 31.04 | - | 0.040 | 8.3 |
| HexPlane[5][1] | 31.71 | 0.014 | 0.075 | - |
| 4DGS-Realtime[47]† | 32.01 | 0.014 | - | 114 |
| Spacetime-Gs[18]† | 32.05 | 0.026 | 0.044 | 140 |
| 4DGS[45]† | 31.15 | 0.016 | 0.049 | 30 |
| Ours | 32.15 | 0.014 | 0.044 | 40 |

movements and occurrences of object appearance and disappearance. This dataset allows for a comprehensive evaluation of the model's reconstruction capability in complex temporal scenes.

## 6.2 Results

We have employed a variety of evaluation metrics to assess our model. For rendering quality, we utilize PSNR, SSIM, DSSIM, and LPIPS, and for rendering speed, we measure FPS. All evaluation results are averaged across all scenes in the dataset.

For monocular scenes in the D-NeRF dataset, we compare our method against the current state-of-the-art methods[5–7, 9, 31, 45, 47] in the field. The quantitative evaluation results are listed in Tab. 1. The dynamic scene representation method based on NeRF[5–7, 9, 31] struggles with achieving real-time rendering due to the need for dense ray sampling during rendering. However, our method achieves a rendering speed of 182 FPS and exhibits considerable improvement in rendering quality. Compared to existing dynamic Gaussian methods[45, 47], our approach demonstrates superior performance in handling complex temporal scenes and achieves a certain level of enhancement in rendering quality. Qualitative comparison results can be seen in Fig. 4. In the standup scene, our method demonstrates better reconstruction of details (such as facial and hand features).

For multi-view scenes in the Plenoptic Video dataset, our evaluation results are presented in Tab. 2. Our method outperforms all compared methods in rendering quality while maintaining superior

**Table 3: The ablation study results across all scenes in the D-NeRF dataset.**

| Method | PSNR(dB)↑ | SSIM↑ |
|--------|-----------|-------|
| No Scale-aware | 35.61 | 0.98 |
| No temporal prop. | 32.29 | 0.96 |
| No Adaptive Optimization | 35.44 | 0.98 |
| No res-Reg. | 35.01 | 0.98 |
| Full | 36.13 | 0.98 |

rendering speed, achieving real-time rendering. The NeRF-based methods[1, 5, 7, 17, 35, 41] have a significant disadvantage in rendering speed compared to ours. 4DGS[45] based on deformation fields has shortcomings in modeling complex temporal situations such as object appearance and disappearance, which we can effectively address. Although Spacetime-GS[18] achieves higher rendering speeds, it is only applicable to multi-view scenes, whereas our method is suitable for both multi-view and single-view scenarios. A qualitative comparison of rendering quality with Gaussian methods can be seen in Fig. 5. Our approach demonstrates more accurate reconstructions in temporally complex scenes (such as the inverted coffee scene with details on the heads and hands) and richer details (such as the circular decorations on the upper garment and reflections on the clippers in the cut beef scene).

## 6.3 Ablation Study

**Temporal properties of 4D Gaussian.** Each Gaussian primitive in 4D space possesses temporal properties, including temporal position and lifespan. These Gaussian primitives can be arbitrarily distributed across the temporal domain, giving us an advantage over previous deformation-based methods in handling intricate temporal information. To demonstrate this, we fix the temporal position of each point to frame 0 and integrate the 3D position of Gaussian primitives with the sampling time to extract features in the Redisual field. 3 and Fig. 6, show a notable decline in performance when temporal position and lifespan are disregarded.

**Consideration of Spatial Scale in the Residual Field.** In our Residual field, we take into account the size information of each Gaussian, improving the accuracy of the features obtained for each Gaussian and aligning with the splitting of Gaussian primitives. To evaluate this approach, we refrain from encoding the projection area in spatial-only planes and conduct assessments on the D-NeRF dataset. As depicted in Tab. 3 and Fig. 6, not incorporating the size information of Gaussian primitives results in a reduction in the reconstruction quality of dynamic scenes.

**Adaptive Optimization.** To address the imbalance in optimization between dynamic and static regions in the scene, we introduce the Adaptive Optimization strategy. Without this strategy, the reconstruction ability of 4D Gaussians for moving regions would decrease. This is supported by Tab. 3. Without this strategy, Gaussian primitives struggle to distinguish between dynamic and static regions, leading to artifacts as shown in Fig. 6(e).

**Regularization of scaling residuals.** We conducted evaluations without regularization of scaling residuals to validate the effectiveness of this regularization term. Without this constraint, Gaussian primitives overly rely on the Scale-aware Resisual field, neglecting optimization of their own initial attributes. Additionally, it leads

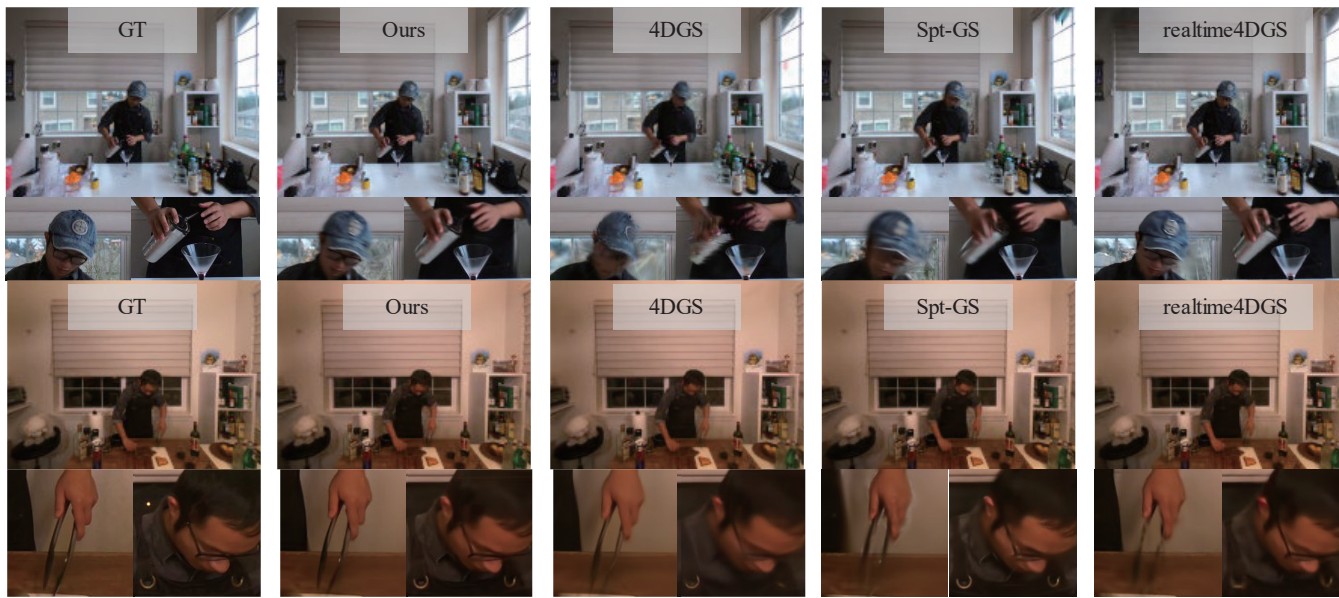

**Figure 5: Qualitative results on coffee martinis and cut roasted beef from the Plenoptic Video dataset**

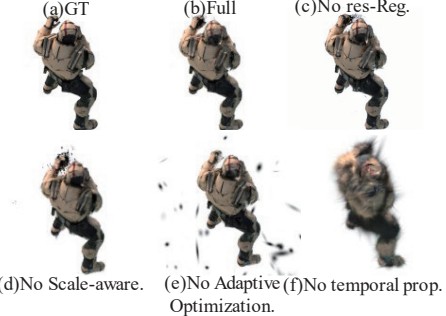

(a)GT  (b)Full  (c)No res-Reg.

(d)No Scale-aware.  (e)No Adaptive  (f)No temporal prop.
Optimization.

**Figure 6: Qualitative results of the ablation study.**

to the utilization of sub-optimized scale information in the Scale-aware Residual field, resulting in a decrease in model performance, as shown in Tab. 3 and Fig. 6.

## 6.4 Limitations

Our method achieves high-quality, real-time reconstruction of temporally complex scenes. However, it has limitations. Firstly, the use of explicit and implicit mixing, such as 4D space plane decomposition and MLP inference, may reduce training speed. Secondly, while applicable to both monocular and multi-view scenarios, accurate pose estimation is required. Thus, in scenarios lacking precise pose calibration, such as handheld smartphone videos, performance may degrade. Enhancing reconstruction quality in such scenarios is a future optimization direction.

## 7 DYNAMIC-STATIC SEGMENTATION

Static primitives are observable throughout the entire observation period, whereas dynamic primitives, aiming to represent the temporal complexity of the scene, are only visible near their temporal positions. Therefore, dynamic primitives have a shorter lifespan, while static primitives have a much longer lifespan. Consequently, we can segment the scene into dynamic and static parts based solely on their lifespans, without any prior knowledge, as illustrated in Fig. 7. Our method accurately segments dynamic foreground elements

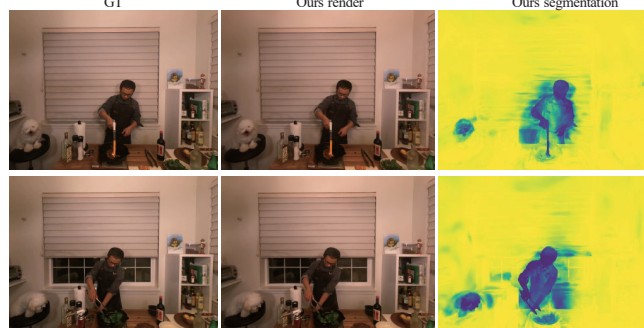

**Figure 7: Segmentation of dynamic and static scenes.**

and dynamic lighting effects (e.g., shadows of people in the image). This aspect also underscores the interpretability of our method.

## 8 CONCLUTION

In this paper, we propose SaRO-GS as a novel approach for representing dynamic scenes, enabling real-time rendering while ensuring high-quality reconstruction, especially in temporally complex scenes. SaRO-GS utilizes a set of 4D Gaussian primitives to represent dynamic scenes and leverages 3D Gaussian Splatting for real-time rendering. Additionally, we propose a Scale-aware Residual Field to encode the region occupied by Gaussian primitives, resulting in more accurate features and align with the self-splitting behavior of Gaussian primitives. Furthermore, we introduce an Adaptive Optimization strategy to enhance the model's ability to reconstruct high-frequency temporal information in dynamic scenes. Experimental results in both monocular and multi-view settings demonstrate that our approach achieves state-of-the-art rendering quality while enabling real-time rendering.

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
