# OpenReview forum: "4D Gaussian Splatting with Scale-aware Residual Field and Adaptive Optimization for Real-time rendering of temporally complex dynamic scenes"
_acmmm.org/ACMMM/2024/Conference — MM2024 Oral_

### Official Review · Reviewer_i357 · 2024-05-21

**Rating:** 5
**Confidence:** 3

**Summary:**

The paper proposes SaRO-GS as a dynamic scene representation for real-time view synthesis of dynamic scenes. The method uses a Gaussian primitive-based representation and optimizes the Gaussians in 4D space. To handle temporally complex dynamic scenes, SaRO-GS introduces a Scale-aware Residual Field, which can produce the lifespan and temporal offsets of each Gaussian. In addition, the paper presents an Adaptive Optimization Schedule to adaptively choose the densify gradient threshold and learning rate of each Gaussian. Experiments are conducted on both monocular and multi-view datasets.

**Strengths:**

1. The paper is well-structured and easy to follow.
2. For the monocular D-NeRF dataset, the proposed method outperforms prior arts such as 4DGS and 4DGS-Realtime by a noticeable margin in rendering quality. The rendering speed is also faster than 4DGS.
3. The paper conducts a detailed ablation study on the proposed components, which demonstrates the effectiveness of each component.

**Limitations:**

1. The idea of Eq. 9-10 is highly related to the temporal opacity proposed in [18]. Hence, it is suggested to reference [18] at these paragraphs.
2. Line 497-499 mention that the size of Gaussian primitives impacts their projection only within the spatial-only planes. However, from my understanding, the size should also impact the projection on the spatiotemporal planes. For example, s_x and \sigma_i can impact the projection on the x-t plane.
3. In Table 2, NeRFPlayer, HyperReel and Spacetime-Gs use different DSSIM computation setting than the other methods, which will lead to higher DSSIM value. Therefore, the paper should mention that it is unfair to directly compare their DSSIM value with the others'. [18] provides a description on the different DSSIM computation settings, and reports the DSSIM metrics of Spacetime-Gs for each computation setting.
4. In the supplementary video, there is flickering near fast-moving objects (e.g., the cooking torch).
5. Missing references to some advances in view synthesis:
"Masked Space-Time Hash Encoding for Efficient Dynamic Scene Reconstruction", "Zip-nerf: Anti-aliased grid-based neural radiance fields", "Neurbf: A neural fields representation with adaptive radial basis functions"

**Suitability:**

3

---

### Official Review · Reviewer_nJVf · 2024-05-24

**Rating:** 6
**Confidence:** 1

**Summary:**

This paper proposes a novel Gaussian-based method to reconstruct dynamic scenes from video sequences. They introduce a Scale-aware Residual Field to handle temporally complex dynamic scenes and propose an Adaptive Optimization schedule to expedite the reconstruction of dynamic regions. Based on their design, the method exhibits high-quality reconstruction and high-efficient rendering.

**Strengths:**

They achieve high-quality reconstruction and real-time rendering simultaneously.

The analysis and design of scale-aware residual field and adaptive optimization, including the spatial projection and leveraging different optimizations for different gaussians, are convincing.

Exhaustive experiments demonstrate its superiority compared to previous methods in terms of reconstruction quality.

The exposition is correct, and the technique can be reproduced.

The writing is clear.

**Limitations:**

The adaptive optimization seems to be capable of accelerating the training, but there is no experiemnt to validate this.

I am not familiar with this field and I cannot find severe limitation of this paper.

**Suitability:**

3

---

### Official Review · Reviewer_CAfd · 2024-05-24

**Rating:** 5
**Confidence:** 3

**Summary:**

This article focuses on the task of reconstructing dynamic scenes in video sequences, with a focus on tracking the appearance and disappearance of objects. Lifespan is introduced as a control to better represent the Gaussian function of objects at any moment.

**Strengths:**

The lifespan proposed in this paper is innovative. At the same time, judging from the pictures and tables in the experiment, this paper has certain advantages over other algorithms in terms of effectiveness.

**Limitations:**

1. From the perspective of the effect, it seems that the introduction of the lifespan parameter has caused shaking and flickering in the video.
2. Will it cause a significant increase in the number of parameters, for example, an object only appears briefly, but its corresponding Gaussian function remains unchanged.

**Suitability:**

3

---

### Official Review · Reviewer_vBfU · 2024-05-25

**Rating:** 6
**Confidence:** 3

**Summary:**

This paper proposes a novel 4D-GS method to represent dynamic scenes using a scale-aware residual field and an adaptive optimization schedule. The research gaps and proposed solutions are well-presented, and experiments on both monocular and multi-view scenarios demonstrate the effectiveness and robustness of the method.

**Strengths:**

The paper is well written and clearly presents their ideas
Their approach is very novel and effectiveness
Their performance achieves sota performance on public monocular and multi-view datasets

**Limitations:**

1.Does Figure1 (d) provide the comparison on the rendering quality and Speed on D-Nerf dataset ？ If so, the comparison with the 4DGS  seems to be unfair as they are rendered with the solution of 800
2.The lifespan allows to model the disappearance of objects, but I wonder how it handles the sudden appearance of new objects
3.The adaptive optimization schedule is designed based on the temporal properties, why such a design can benefit the performance of performance? What is the motivation behind this?
4.It seems to be unfair to compare the efficiency with different resolutions as can be seen in table 1? Can you explain the reasons for this?
5.Table 2 does not show the sota rendering speed
6.I feel the ablation study section can be improved. Detailed experiments is preferred to validate the effectiveness of each component, starting from the baseline, which does not adopt any proposed strategies

**Suitability:**

3

---

### Meta-Review · Area_Chair_s9B6 · 2024-07-01

**Recommendation:** Accept (Oral)
**Confidence:** 5

**Metareview:**

The paper presents a novel method focused on reconstructing real-time dynamic scenes in video sequences. Experiments are conducted on both monocular and multi-view datasets.

Since the first round of ratings, all the reviewers agreed on the good quality of the paper in terms of writing, novelty of the contribution and convincing analysis and results presented. Only one of the four reviewers is not expert in the topic.
Two reviewers (Reviewer vBfU and i357) raised some concerns (e.g., ablation study quality, flickering in the video submitted as supplementary material and about some choices made by authors) but they have been all successfully addressed in the rebuttal. Authors also presented additional analysis and results in the submitted rebuttal.

Given the above remarks, I believe the paper presents a relevant contribution for ACM Multimedia community. My suggestion is also to have it as an oral presentation.